# Women's lived experiences of preterm birth and neonatal care for premature infants at a tertiary hospital in Ghana: A qualitative study

**Kwame Adu-Bonsaffoh**[1,2,3]*, **Evelyn Tamma**[2], **Adanna Uloaku Nwameme**[4], **Martina Mocking**[3], **Kwabena A. Osman**[5], **Joyce L. Browne**[3]

**1** Department of Obstetrics and Gynaecology, University of Ghana Medical School, Accra, Ghana, **2** Holy Care Specialist Hospital, Accra, Ghana, **3** Julius Global Health, Julius Center for Health Sciences and Primary Care, University Medical Center Utrecht, Utrecht University, Utrecht, The Netherlands, **4** Department of Social and Behavioural Sciences, School of Public Health, University of Ghana, Accra, Ghana, **5** Department of Child Health, University of Ghana Medical School, Accra, Ghana

* kadu-bonsaffoh@ug.edu.gh

**Data Availability Statement:** All relevant data are within the paper and its Supporting information files.

## Abstract

Preterm birth is a leading cause of death in children under five and a major public concern in Ghana. Women's lived experiences of care following preterm birth in clinical setting represents a viable adjunctive measure to improve the quality of care for premature infants. This qualitative study explored the knowledge and experiences of women who have had preterm birth and the associated challenges in caring for premature infants at a tertiary hospital. A qualitative design using in-depth interviews (IDIs) was conducted among women who experienced preterm birth with surviving infants at the Korle-Bu Teaching Hospital in Accra, Ghana. A thematic content analysis using the inductive analytic framework was undertaken using Nvivo. Thirty women participated in the study. We observed substantial variation in women's knowledge on preterm birth: some women demonstrated significant understanding of preterm delivery including its causes such as hypertension in pregnancy, and potential complications including neonatal death whilst others had limited knowledge on the condition. Women reported significant social and financial challenges associated with preterm birth that negatively impacted the quality of postnatal care they received. Admission of preterm infants at the neonatal intensive care unit (NICU) generated enormous psychological and emotional stress on the preterm mothers due to uncertainty associated with the prognosis of their babies, health system challenges and increased cost. Context-specific recommendations to improve the quality of care for prematurely born infants were provided by the affected mothers and include urgent need to expand the National Health Insurance Scheme (NHIS) coverage and more antenatal health education on preterm birth. Mothers of premature infants experienced varied unanticipated challenges during the care for their babies within the hospital setting. While knowledge of preterm birth seems adequate among women, there was a significant gap in the women's expectations of the challenges associated with the care of premature infants of which the majority experience psychosocial, economic and emotional impact.

**Funding:** This study was supported by the UMC Utrecht Global Health Fellowship in the form of an award to KAB. The funders had no role in study design, data collection and analysis, decision to publish, or preparation of the manuscript.

**Competing interests:** The authors have declared that no competing interests exist.

## Introduction

Preterm birth (PTB) is the leading cause of death in children under five worldwide [1] and a major public health challenge worldwide [2–4]. The World Health Organization defines preterm birth as a childbirth that occurs before 37 weeks of gestation or less than 259 days from the first day of the last menstrual period [2]. Globally, about 15 million babies are born premature every year and more than one million of them die immediately after their birth [2]. Although preterm birth is a global phenomenon, the main burden of the resulting prematurity is far higher in low and middle- income countries (LMIC), with approximately 80% occurring in sub-Saharan Africa and South Asia [2, 5]. In Ghana, a country in sub-Saharan Africa, the second leading cause of death among children under five is prematurity with a national rate of 14.5% [6]. Annually, more than 100,000 premature babies are born in the country with direct complications of preterm birth contributing to approximately 8,200 under five child deaths [6].

In the effort to achieve the 2030 Sustainable Development Goal (SDG) 3: to eliminate preventable neonatal and under-5 mortalities, the burden of preterm births needs a global attention. Preterm babies are at risk of major health complications such as respiratory distress syndrome, sepsis, feeding difficulties, vision and hearing impairment, learning disabilities and chronic lung disease [7]. Preterm infants usually require specialized care with prolonged hospitalization resulting in an increased psychological stress and financial burden on affected families [8, 9]. The etiology of spontaneous PTB still remains unclear and provider-initiated preterm phenotype has varied causal associations [7]. Known common risk factors for preterm birth include multiple gestation, infection and chronic diseases such as diabetes and hypertension (pre-eclampsia), preterm premature rupture of membrane (PPROM), advanced maternal age and low socio-economic status [7, 10, 11].

Recently, the WHO highlighted evidence-based interventions to improve preterm birth outcomes as part of the global effort of reducing child mortality [12]. A recent systematic review on evidence-based intervention to reduce mortality among preterm and low birth weight neonates identified additional useful interventions such as cord and skin cleansing with chlorhexidine [13]. In addition to the recommendations that focus on the clinical provision of care, a better understanding of the plight of the preterm mothers and their lived experiences of care at the facility level can inform optimal implementation of these interventions. While some research exists from various settings that indicate that preterm birth women experience major challenges across the continuum of antenatal, intrapartum, postpartum and newborn care, studies from sub-Saharan Africa are limited [14].

In Ghana, there is limited research on preterm birth in general resulting in inadequate evidence to inform policy optimally [14]. More specifically, there is limited qualitative research to better understand the social, psychological and clinical challenges associated with preterm birth and the care for the preterm infants. Given the high prevalence of preterm birth in Ghana accompanied by increased maternal psychological and financial burden, the lived experiences of these mothers are vital in shaping the quality of care for the mothers and their premature infants. There is evidence that an individual's knowledge about a health condition can influence early seeking of medical care, and can prevent (further) complications [15], improve compliance to treatment [16] and maternal and newborn's health outcomes and care experiences. Therefore, this paper explores the knowledge and lived experiences of women who have experienced preterm birth and the associated challenges in caring for premature infants at a tertiary hospital.

## Methods

### Study design and sites

This was a qualitative phenomenological study, using in-depth interviews (IDIs), conducted to gain comprehensive understanding of the women's knowledge on preterm birth including its causes and associated complications and their lived experiences of care. The study was carried out at the Maternity Unit of the Korle-Bu Teaching Hospital (KBTH) in Accra from August 2019 to January 2020. KBTH is the largest teaching hospital in the country and serves as a tertiary referral center for the Greater Accra Region and its environs. Annually, the KBTH conducts approximately 10,000 deliveries and nearly one in five pregnant women experience preterm birth [10]. The preterm birth rate of approximately 20% has been reported with about 50% NICU admission rate [10]. The hospital has a moderately well-equipped neonatal intensive care unit situated in the Maternity unit, managing preterm infants and neonates requiring intensive care services.

### Study population

Women of 18 years and above who delivered preterm (less than 37 completed weeks of pregnancy) at the health facility, and provided informed consent were eligible for participation. We included preterm births with livebirths from spontaneous onset of labor (including preterm pre-labor rupture of membranes) or provider-initiated (medically indicated) phenotype including caesarean deliveries for maternal or fetal indications. Women were excluded if they had multiple pregnancies due to their increased risk for preterm birth and pregnancies that ended in stillbirth. We excluded the women with stillbirths because we were primarily interested in understanding the experiences of caring for preterm neonates especially at the neonatal intensive unit.

### Recruitment, sampling and data collection

Purposive sampling was used to recruit participants into the study with the aim of achieving varied obstetric history, maternal age and socio-economic characteristics. Purposive sampling is a non-probability sampling involving initial identification and selection of study participants of interest who are proficient and optimally informed concerning the phenomenon being studied [17, 18]. In this study, the Research Assistant (RA) identified women who met the inclusion criteria. These women were then approached and the study protocol was explained to them individually by a trained research assistant (ET). The RA had Master's degree in Statistics and was a non-health worker with considerable experience in conducting IDIs. Mothers who consented were then scheduled for an interview immediately after their discharge from the hospital. The IDIs were conducted in a quiet designated room at the hospital. All the respondents provided written informed consent before the interviews. Recruitment of participants continued until no new themes emerged from the data (data saturation was reached). The IDIs were carried out in English, Twi or Ga (two local Ghanaian dialects) and audio recorded. The interviews lasted between 17 to 50 minutes. We used a semi-structured interview guide (S1 File) with varied questions and appropriate probing was done during the interviews.

### Ethical consideration

The study protocol was approved by the Ethical and Protocol Review Committee of the College of Health Sciences University of Ghana (Protocol ID: CHS-EtM.4-P1.2/2017-2017). Written informed consent was granted by all participants prior to collection of data. We ensured

anonymity by non-inclusion of any identifiable information on the women included in the study.

### Data management and analysis

Transcription of the interviews started soon after the commencement of data collection. Back-to-back translation of all IDIs from Twi or Ga (Ghanaian dialects) into English were done and ET checked all transcripts for accuracy and completeness. A code book was developed from the review of the transcripts and themes that emerged from the data were identified. The coding was done by ET and the principal investigator (KAB), who is an obstetrician/gynecologist at KBTH. Discrepancies were resolved via discussion between KAB and ET. NVivo version 12 software was used for the coding and analysis. In this study, the data analysis was undertaken based on the thematic content by employing the inductive analytic framework where the themes were based on recursive reading of the transcripts with minimal theoretical background influence [19]. Triangulation of the study findings was achieved via the inclusion of different case-mix comprising women with both spontaneous and provider-initiated preterm births. The authors of this study had different but relevant specialties including obstetricians, epidemiologists, social scientist, and pediatricians. The convergence of different researchers with related research interest created the needed reflexivity and interpretation of the study findings. In writing this paper, the Consolidated criteria for reporting qualitative research (COREQ) were employed as a guide [20].

### Results

A total of 50 women were invited for the in-depth interviews, out of which 30 provided informed consent and participated. The reasons given by those who declined participation in the interview included: the need to follow up on laboratory results of their babies and did not have time; some mothers had to return home to attend to their other young children; or they were emotionally stressed from their baby's admission at NICU and were therefore not interested in the study.

Most of the women had Junior High School level of education (n = 13, 43.3%) and were married or cohabiting (n = 19, 63.3%). Half of the women were petty traders (non-professionals) and between the ages of 20–30 years (n = 15, 50%). Majority of the women had 1–4 previous deliveries (n = 29, 96.7%) and most had no history of previous preterm birth (n = 28, 93.3%) (Table 1).

The major themes that emerged from the narratives of women who had recently experienced preterm birth were:

1. Women's knowledge on preterm birth

2. Financial burden of preterm birth to families

3. Mothers' experiences of neonatal intensive care unit admission

4. Recommendations to improve the quality of care

**1. Women's knowledge on preterm birth**. Women's knowledge on preterm birth was a major theme that emerged and mixed findings were reported concerning the causes and complications. An important sub-theme that emerged from the interviews related to the causes of preterm delivery and the study participants described various conditions associated with preterm birth. Hypertension in pregnancy, which is locally referred to as "BP", was mentioned as the main cause of preterm delivery. Stress-related issues and physical activities

**Table 1. Socio-demographic characteristics of women who experienced preterm birth.**

| Variable | Number (n) | Percentage (%) |
|---|---|---|
| **Age (years)** | | |
| <20 | 2 | 6.7 |
| 20–29 | 15 | 50.0 |
| 30–39 | 11 | 36.7 |
| ≥40 | 2 | 6.7 |
| **Marital status** | | |
| Single | 11 | 36.7 |
| Married/co-habiting | 19 | 63.3 |
| **Education** | | |
| None/primary | 2 | 6.7 |
| Junior High School | 13 | 43.3 |
| Senior High School | 11 | 36.7 |
| Tertiary | 4 | 13.3 |
| **Number of previous births** | | |
| 0 | 1 | 3.3 |
| 1 | 14 | 46.7 |
| 2 | 6 | 20.0 |
| ≥3 | 9 | 30.0 |
| **Religion** | | |
| Christian | 29 | 96.7 |
| Muslim | 1 | 3.3 |
| **Ethnicity** | | |
| Akan | 13 | 43.3 |
| Ewe | 3 | 10.0 |
| Ga | 9 | 30.0 |
| Other | 5 | 16.7 |
| **Occupation** | | |
| Unemployed | 3 | 10.0 |
| Hairdresser | 5 | 16.7 |
| Trader | 15 | 50.0 |
| Seamstress | 3 | 10.0 |
| Others | 4 | 13.3 |
| **Previous preterm birth** | | |
| Yes | 2 | 6.7 |
| No | 28 | 93.3 |

such as frequent bending down were also cited by some respondents as causes of preterm birth.

> "*They said a woman could deliver before her time and that could be as a result of BP, or if the baby is not well positioned that could also make the baby come before the time.*"

(*Married, 38 years*)

> "*High BP can cause it. High pressure can make you deliver preterm.*"

(*Married, 28 years*)

"*If you bend down too much, over stressed, things like that can cause it or thinking or you have some marriage problems, plenty things like that.*"

(*Married, 30 years*)

Most of the participants had adequate knowledge about the neonatal complications associated with preterm birth, and mothers mentioned complications associated with preterm birth such as infant death, poor physical health and infection.

"*You may lose the baby. Some of the babies survive but they don't have the full strength to work or do something, may be physical activities.*"

(*Single, 25 years*)

"*When you deliver the baby too early and in case the hospital where you delivered does not have an incubator, the baby can get an infection and can die instantly.*"

(*Married, 28 years*)

Some participants indicated the possibility of poor development or formation of some organs in the infants (congenital anomalies) such as the eyes or ears. This is evidence suggesting that some of the mothers were well-informed about preterm birth and its complications.

"*May be a part of the body may not be well formed. May be the eye or ear.*"

(*Trader, 31 years*)

On the other hand, some women had limited knowledge on preterm birth and its causes. There were significant misconceptions especially nutritional associations with preterm birth. Health education and more research into the causes of preterm birth were recommended to improve women's understanding on the condition.

"*If we had been educated may be, I wouldn't have eaten what I wasn't supposed to. Be it the diet or thinking, I don't know what caused it but if they can carry out research into it to find out the cause and are able to educate us, I'm sure these things (preterm deliveries) wouldn't go on.*"

(*Cohabiting, 27 years*)

**2. Financial burden of preterm birth to families**. Another important theme that emerged was the effect of hospital admission on participants and their families. Financial constraints greatly affected the preterm mothers, and their partners had to make extra efforts to raise the additional funds needed to pay hospital bills. The plans of families were also disrupted and in some instances some husbands could not go to work as they had to visit their wives in the hospital frequently. Following discharge, some women could not afford the cost of treatment and were not able leave the hospital until the bill was paid. Such mothers were frequently detained further in the hospital until all hospital bills were settled.

"*I have been discharged and the bill is 350 Ghana cedi [60 USD]and my husband hasn't got the money. He has gone round and hasn't been able to raise the money.*"

(*Married, 33 years*)

"*I have been discharged now and my husband went to find out how much my bill is. They have given him the amount so he has gone out to look for the money so that he can get me discharged.*"

(*Married, 28 years*)

*Since my admission, my husband has not gone to work for three days. He can't go to work because it is just the two of us. He is always here seeing to the purchase of my medication. It prevents you from making money and it takes up all your time. It has messed everything up.*"

(*Married, 28 years*)

The national health insurance scheme (NHIS) provided minimal coverage for the facility bills leaving the preterm mothers and their families with the burden of paying most of the bills out-of-pocket. Some of the participants asserted that the insurance scheme covers only the less costly items in their medical bill and the more expensive ones are paid by the mothers themselves.

"*Since I came here, we bought all my drugs except the one that you insert [rectally]. My husband said he bought all the other drugs.*"

(*Married, 28 years*)

"*Just the things that are not expensive—that is what the NHIS will cover.*"

(*Married, 31 years*)

Although there is some NHIS coverage for neonatal care, most of the women had to pay for a significant portion of the medical bills related to oxygen therapy, laboratory test and medications for the preterm infants. The women's narratives indicated the entrenchment of cost-sharing between the NHIS and parents. In some situations, the mothers verified the treatment provided to their babies before making payment for their hospital bill.

"*I'm paying 405 Ghana cedis [68 USD] after receiving support from insurance.*"

(*Married, 31 years*)

"*All I saw was ward fee and oxygen. When I visited the baby on the second day I didn't see any oxygen being given to him so I asked her why I was charged for oxygen and she said the baby was given. She brought the confirmation which had been recorded that the baby was on oxygen for 24 hours.*"

(*Married, 38 years*)

"*I paid it myself. I bought the drugs and paid for the labs. Insurance covered some and I also paid for some.*"

(*Married, 22 years*)

**3. Mothers' experiences of neonatal intensive care unit (NICU) admission**. Most of the prematurely born babies were admitted to NICU. Generally, mothers of preterm infants are informed about the need for admission of their preterm infants to the neonatal intensive care unit (NICU). Although the narratives indicate that the mothers are informed about the

need for NICU admission, no comprehensive counseling and education on expectations are provided.

> "*So all I heard was the baby's crying and they came to show her to me saying 'you had a girl' and I said 'okay'. Then they said she was small so they are sending her to NICU.*"
>
> (*Cohabiting, 24 years*)

> "*They showed the baby to me after he was taken out: [..] 'this is your baby, but because his time wasn't yet up, they will send him to NICU so that he can regain his strength'.*"
>
> (*Married, 30 years*)

The NICU admissions had major psychological and emotional effect on the mothers, mainly due to the uncertainty of their preterm babies' survival. Majority were not happy about the fact that they were separated from their babies and not knowing what is happening to their babies. Women who had previous experiences of delivering preterm and losing their babies at NICU were terrified of potentially experiencing infant demise for the second time.

> "*I feel very sad, [..] there is no mother who would feel happy when she is somewhere else and her baby [is] at another end.*"
>
> (*Married, 28 years*)

> "*I was scared when they took the baby there [NICU] because my first pregnancy. That pregnancy was [ended at] about 26 weeks. They took the baby there and the next day the baby passed on. So this one I was so scared. I was praying to God that, 'please this one spare me'.*"
>
> (*Single, 27 years*)

> "*You've delivered so your baby should be laying by you. So I don't like going there [NICU] because leaving him behind is painful.*"
>
> (*Married, 35 years*)

The burden of having to visit their babies several times at NICU was very challenging for most mothers especially those who have had a caesarean section. Mothers who lived far from the hospital had to spend the entire day at the hospital as it was challenging for them to travel back and forth. In addition, some mothers spent long hours at the laboratory to wait for reports of their babies' laboratory tests. This disrupted their expectations of motherhood in the postpartum period and had an enormous impact on their finances and emotions.

> "*I have had CS and having to walk up and down is very worrying for me. When the baby is on admission the labs and all that are a lot, which make the cost very high. You are given a new lab test for the baby every day.*"
>
> (*Married, 38 years*)

> "*I'm disturbed* (*Respondent breaks down in tears*). *I wish I had delivered my baby at 9 months. I live far from here [the hospital] and I come here and see the baby three times a day: If you come in the morning you have to wait till evening because you have to go and see him in the afternoon and evening. Everything has been messed up so I'm very worried.*"

(*Married, 28 years*)

"*It took a long time because they [lab technicians] said I couldn't get the lab results [of the baby]. So I pleaded with them and told them I had been operated on so I couldn't go and come back so I had to sit down. So I really kept long. I got there at 4 [pm] but when I was leaving it was around 7 [pm].*"

(*Married, 37 years*)

**4. Women's recommendations to improve the quality of care**. The preterm birth mothers explained how overwhelming the medical bills for them and their babies were. This resulted in some of them absconding from the hospital without paying their bills. They indicated that the NHIS does not cover the cost of most of the medications and laboratory tests needed by the preterm neonates. They therefore urgently recommended wider NHIS coverage of services to ease the financial burden associated with preterm delivery and neonatal care for premature infants.

"*The insurance should be able to cover majority of the cost. When you are discharged you pay a lot and the insurance only covers a small portion. So, the government should ensure that it [NHIS] covers majority of the labs for us*"

(*Married, 38 years*)

"*Three days and the drugs are almost 700 Ghana cedis [115 USD] plus. And what about the baby in the incubator? Staying here for long [myself] and the baby too being on admission, when they discharge her and the money is a lot, that brings about a huge burden. So we will plead with the government for any form of support so that when it [preterm birth] happens (..) they can take care of the bills for us.*"

(*Married, 28 years*)

Some mothers emphasized that they do not usually have an idea about the amount of money they will pay for their infants' admission. This makes them inadequately prepared to pay their bills and result in psychological stress for them and their families. One mother speculated that her preterm infant might be on admission at neonatal intensive care for over a month and guessed she might pay a lot of money following discharge. She estimated the medical bill to be GHC 1500 (250 USD; 1USD was approximately 6GHC). The rate She pleaded with the government to come to their aid.

"*I don't know the baby's bill and I'm also here. They could come and ask me to pay the bill when the baby is 1 month 2 weeks or something. I might be asked to pay GHC1000 or GHC1500 [160–250 USD], when that happens you are disturbed so I will plead with the government to help in this aspect.*"

(*Married, 28 years*)

Health education on preterm delivery with emphasis on treatment was also recommended by majority of the respondents. Also, timely interventions such as giving medications that will help mothers and babies was requested.

"*If it depends on the medicines that we have being taking, they should educate us well. Maybe if you take too much of this drug, it can cause this or maybe when you are too stressed it can cause this. If they give us these forms of education, we will be aware and careful so that all these will end and not happen again.*"

(*Married, 30 years*)

## Discussion

In this qualitative study we explored the knowledge and lived experiences of women who experienced preterm birth and the associated challenges in caring for premature infants. There were mixed findings concerning women's knowledge on preterm birth with some mothers displaying significant understanding including the causes and potential complications. We found that women who recently experienced preterm birth faced significant social and financial challenges which negatively impacted on their experience of care. Admission of preterm infants to the neonatal intensive care unit also generated enormous psychological and emotional stress on the affected mothers due to uncertainty relating to the prognosis of their babies and the unexpected health system challenges including increased cost of care. Important recommendations to improve the experience of care for prematurely born infants were proffered by the affected mothers including the urgent need for expansion of the NHIS coverage and more client education.

Women demonstrated substantial knowledge about the causes of preterm birth and associated complications such as neonatal death. However, the mothers were not adequately prepared for the challenges of neonatal admission which resulted in substantial emotional stress and negative experiences during the postnatal period, consistent with observations by Brandon et al [21] in the United States. An important challenge faced by the mothers was the stress of multiple visits to see their babies in the NICU, especially for women who delivered by cesarean section. Modification of NICU facilities to align with WHO standards for improving quality of care [22] and provision of spaces or family rooms for mothers with preterm infants may improve bonding, direct hand-on care and reduce the stressful postnatal experiences by the mothers [23]. This will provide parents the opportunity, motivation and empowerment to offer parental support to their babies including Kangaroo mother care, as well as other care practices. There is an urgent need for improved ANC education to create more awareness about preterm birth and available neonatal care services for premature infants to adequately prepare prospective mothers for the unanticipated postnatal challenges. Similarly, improvement in the physical and logistic set up of the hospital could also address the long waiting hours at the laboratory centers for babies' laboratory tests, which lead to extreme distress especially for the cesarean mothers. For example, with automatic direct or electronic sharing of laboratory results to the care providers can lessen the burden on the already stressed mothers who are recovering from childbirth. The electronic medical records system is currently being implemented in the hospital, and when fully established, can help send laboratory results directly to doctors to relieve mothers of this burden.

NICU admission had significant psychological and emotional effect on the mothers. This was due to uncertainty about the survival prognosis for the preterm infants perceived by the mothers, separation of the babies from their mothers resulting in poor bonding, financial worries, and other causes. There is evidence that mothers of preterm infants are at greater risk of postpartum depression (PPD) than mothers of term infants [24]. In the Netherlands, Blom et al reported that mothers who experienced more than two perinatal complications such as

emergency cesarean delivery, hospitalization of the baby at birth, pre-eclampsia, and hospitalization during pregnancy had increased risk of developing PPD [25]. Similar maternal emotional concerns have been reported in other studies [21, 26]. In Malawi, Gondwe et al reported that mothers of preterm infants experienced severe distress symptoms and cesarean section was associated with worse postpartum experiences [26]. In this study, women who had a previous experience of neonatal death following preterm birth were terrified of losing their babies again. These findings highlight the urgent need for integration of psychological support for all women who experience preterm birth to improve their postnatal experiences of care, and particularly for women and families who experience bereavement.

Similarly, majority of women expressed how their babies were immediately separated from them after birth resulting in impaired mother-baby bonding. Recent evidence determined that early parental interaction reduces the risk of late PPD but has no influence on early maternal adjustment or late mother-infant relationship [27]. In a related study, immediate separation of mothers and their babies after birth created challenges in women conceiving themselves and functioning optimally as mothers [28, 29]. This phenomenon of "powerless responsibility" describes the situation when mothers are knowledgeable and yet powerless to influence their infants' feeding which negatively impacts on their instinctive mother-care [28]. However, this challenge may be ameliorated via effective client-centered communication and care, including counselling of the affected mothers. WHO recommends mother-to-baby bonding should commence immediately after childbirth [22]. However, in preterm birth, most of the newborns are immediately transferred to the neonatal care unit frequently leaving the mother isolated, powerless and emotionally challenged as their birth expectations of motherhood are curtailed with substantial insecurity about the health status of their babies. Appropriate maternal education, optimal counseling, effective provider-client communication and supportive physical environment are important adjunctive measures for allaying the emotional and psychological influences of quality of care. Allowing women to be close to their infants, even during NICU admission is recommended to enhance mother-infant bonding and parental involvement in the care for the baby.

Although maternity care services under the NHIS are supposed to be free in Ghana [30, 31] the coverage was reported by women to be only minimal. They narrated instances where they had to pay for medications and laboratory tests that were not covered by the NHIS resulting in significant financial constraints on the family. The financial consequences and out-of-pocket expenditures have also been reported by other studies [31, 32]. A recent study in Ghana explored parental cost for hospitalized neonatal services and reported that irrespective of health insurance status, medical-associated costs accounted for approximately 66% of out-of-pocket payments and the mean out-of-pocket expenditure was about $148 for preterm or low birth weight [33]. In Ghana, the mean household size is approximately 6.5 and the average household annual income is approximately GH₵33,937 (6,650 USD) with monthly household income of GH₵ 2,830 (470 USD) [34]. Approximately 10% of parental annual income can be expended on acute care for premature or low birth weight infants [33]. In these circumstances, financial constraints can potentially delay the care process and result in suboptimal care. Most of the women therefore recommended for measures to increase the coverage of the NHIS by the government to ease the financial burden of the affected mothers, as others have also called for in other studies [35]. Similar national health insurance challenges have been reported in other LMICs including inadequate implementation and coverage [36].

In the Ghanaian context, family members play a major role in caring for their relatives on hospital admission including contribution to the payment of hospital bills, chasing of laboratory reports from laboratory centers, running to purchase medications and provision of psychological/social support. There is evidence that provision of social support by family relations

improve the physical and psychological health of the mothers and enhance attainment of optimal maternal role in the postpartum period [37]. On the other hand, financial constraint in settling hospital bills is a major challenge for most mother and their families. Although the NHIS is expected to cover maternal health services (free delivery policy through the NHIS) [38], there is evidence that women and their relatives pay considerable proportion of their medical bill via out-of-pocket [30, 39]. In addition to the burden of financial constraints or poverty, there are real-time health system challenges that preclude provision of comprehensive care for mother and their babies without the support from family members. To reduce unnecessary health system delays in provision of care and reduce the high perinatal morbidity in the country, an urgent paradigm shift in re-structuring the health system is needed to minimize significant dependance of family members especially for hospitalized mothers and preterm infants. An effective collaboration between the health system and NHIS is required to independently provide comprehensive clinical care to mothers without over-reliance on financial or physical support from families. To this direction, effective health system and NHIS financing or management with realistic costings for medical services is urgently recommended as a context specific intervention to improve maternal and newborn care in the country.

## Strength and limitations

The main strength of this study relates to the qualitative design which provided the opportunity for the mothers of prematurely born infants to openly share their lived experiences of care emphasizing on the challenges and key recommendations for improvement. The conduct of the interviews after the mothers had been discharged also enabled the mothers to freely discuss the challenges they encountered without fear of not receiving optimal care.

The limitations of the study include the single tertiary hospital where the study was conducted compared to a multicenter study which would have provided a more comprehensive and varied descriptions of women's experiences of preterm birth in Ghana. Also, the use of only one interviewer could have introduced some form of bias: the interviewer was however experienced in qualitative data collection thus reducing the potential for significant bias. Another important limitation was the use of in-depth interviews only for the data collection without inclusion of focused group discussions (FGDs). The non-inclusion of FGDs may have influenced the triangulation and the applicability of the findings. FGDs would have generated diverse women's experiences of preterm birth in a more interactive manner. In addition, key informant interviews of health workers and key policy makers were not included. The perspectives of health workers and policy makers would have provided a more comprehensive understanding of the provision and experience of care related to preterm birth. Notwithstanding, the findings of this study provides significant insight into the clinical and postnatal experiences of women who deliver preterm in a typical tertiary hospital in Ghana.

## Conclusion

This study has highlighted the lived experiences of women who had encountered preterm birth and the associated postnatal challenges in caring for their premature infants. There were mixed findings concerning women's knowledge of preterm birth, with significant gaps in women's preparedness for postnatal expectations and care for preterm infants. Mothers of preterm infants faced significant social and financial challenges which negatively influenced on the quality of postnatal care. Also, admission of preterm infants to the neonatal intensive care unit created in enormous psychological and emotional stress on the affected mothers due to uncertainty relating to the prognosis of their babies. The affected mothers recommended an urgent need for expansion of the NHIS coverage for neonatal care for preterm babies and

more health education to improve the quality of care for prematurely born infants and women's postnatal experiences. Further research context-related interventions to prevent preterm birth and improve the care for prematurely born infants is recommended to enhance positive postnatal care.

## Supporting information

**S1 File. Interview guide for this qualitative study.**
(DOCX)

## Acknowledgments

We are grateful to the staff of the Department of Obstetrics and Gynecology of Korle-Bu Teaching Hospital for their cooperation and help during the data collection. Our profound gratitude also goes to all the women who consented to take part in the study.

## Author Contributions

**Conceptualization:** Kwame Adu-Bonsaffoh, Evelyn Tamma, Adanna Uloaku Nwameme, Martina Mocking, Kwabena A. Osman, Joyce L. Browne.

**Data curation:** Kwame Adu-Bonsaffoh, Evelyn Tamma, Martina Mocking, Kwabena A. Osman, Joyce L. Browne.

**Formal analysis:** Kwame Adu-Bonsaffoh, Evelyn Tamma, Adanna Uloaku Nwameme.

**Investigation:** Kwame Adu-Bonsaffoh, Evelyn Tamma, Adanna Uloaku Nwameme, Martina Mocking, Kwabena A. Osman, Joyce L. Browne.

**Methodology:** Kwame Adu-Bonsaffoh, Evelyn Tamma, Adanna Uloaku Nwameme, Martina Mocking, Kwabena A. Osman, Joyce L. Browne.

**Project administration:** Kwame Adu-Bonsaffoh, Evelyn Tamma, Adanna Uloaku Nwameme, Martina Mocking, Joyce L. Browne.

**Resources:** Kwame Adu-Bonsaffoh, Evelyn Tamma, Martina Mocking, Kwabena A. Osman, Joyce L. Browne.

**Supervision:** Kwame Adu-Bonsaffoh, Adanna Uloaku Nwameme, Martina Mocking, Kwabena A. Osman, Joyce L. Browne.

**Visualization:** Kwame Adu-Bonsaffoh.

**Writing – original draft:** Kwame Adu-Bonsaffoh, Evelyn Tamma.

**Writing – review & editing:** Kwame Adu-Bonsaffoh, Evelyn Tamma, Adanna Uloaku Nwameme, Martina Mocking, Kwabena A. Osman, Joyce L. Browne.

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
