## [Decision Letter · Decision Letter 0]

2 Jun 2022

PGPH-D-22-00541

Women’s lived experiences of preterm birth and neonatal care for premature infants at tertiary hospital in Ghana: A qualitative study

Dear Dr. Adu-Bonsaffoh,

Thank you for submitting your manuscript to PLOS Global Public Health. After careful consideration, we feel that it has merit but does not fully meet PLOS Global Public Health’s publication criteria as it currently stands. Therefore, we invite you to submit a revised version of the manuscript that addresses the points raised during the review process.

Please submit your revised manuscript by . If you will need more time than this to complete your revisions, please reply to this message or contact the journal office at globalpubhealth@plos.org. Please include the following items when submitting your revised manuscript:

We look forward to receiving your revised manuscript.

Kind regards,

Rachel Hall-Clifford

Academic Editor

Journal Requirements:

1. In the online submission form, you indicated that "The transcripts for this qualitative study will be made available upon reasonable request from the corresponding author". All PLOS journals now require all data underlying the findings described in their manuscript to be freely available to other researchers, either 1. In a public repository, 2. Within the manuscript itself, or 3. Uploaded as supplementary information.

2. We have noticed that you have uploaded Supporting Information files, but you have not included a list of legends. Please add a full list of legends for your Supporting Information files after the references list. 

Additional Editor Comments (if provided):

Thank you for submitting your manuscript. The work shows promise, but significant revisions are needed to make the manuscript ready for publication. Particular attention should be paid to the care context in Ghana and in developing a theoretical framework to position the results.

Reviewers' comments:

Reviewer's Responses to Questions

**Comments to the Author**

1. Does this manuscript meet PLOS Global Public Health’s publication criteria? Is the manuscript technically sound, and do the data support the conclusions? The manuscript must describe methodologically and ethically rigorous research with conclusions that are appropriately drawn based on the data presented.

Reviewer #1: Yes

Reviewer #2: Yes

2. Has the statistical analysis been performed appropriately and rigorously?

Reviewer #1: N/A

Reviewer #2: Yes

3. Have the authors made all data underlying the findings in their manuscript fully available (please refer to the Data Availability Statement at the start of the manuscript PDF file)?

Reviewer #1: Yes

Reviewer #2: Yes

4. Is the manuscript presented in an intelligible fashion and written in standard English?

Reviewer #1: Yes

Reviewer #2: No

5. Review Comments to the Author

Reviewer #1: The manuscript is generally well written and touches on the subject of prematurity which is an old problem with renewed concern following the increasing use of assisted reproductive technology in developing countries. However, these minor changes need to be effected.

1. Title: There should be a inserted before the tertiary.

2. Abstract: well written

3. Introduction: well written

4. Methods: Doing focus group discussions for the mothers to further explore their experiences in a group setting would have further enhanced the quality of your work. As I understand, maternal and new born health are covered under the Ghana NHIS. How come mothers are still complaining of financial burden? Key informant interviews of healthcare personnel and key policy makers could have helped unravel some of these challenges. If it is possible to be done, the authors should consider doing so, otherwise this should be captured also as a limitation in the study.

5. The result is well written.

6. The authors must get more literature to explain why cost should be a great burden in a country where maternal and neonatal health is reported to be fully covered on the insurance. Is this observation reported in other countries? Instead of recommending for a full coverage, the authors should rather call for a probe as to why the mismatch between policy and practice and use other best practices elsewhere to suggest a workable solution.

7. General comments: Kindly read through the work thoroughly and correct all typos, spacing and complete your sentences. For eg line 90 is not completed.

8. No commendation for the NICU staff. Are they part of the Obgyn dept?

Reviewer #2: This research reports the hospital experience of mothers with preterm birth in Ghana. Using qualitative methodology, four themes were identified: knowledge of the risk of preterm birth and its complications, experience of newborn admission to the neonatal intensive care unit, financial burden to families of preterm birth, and recommendations to improve the quality of care.

The manuscripts reports the universal experiences of families after preterm birth and NICU admission, regardless of location of birth: fear regarding uncertainty of outcome, maternal emotional distress at separation from the newborn, and financial burden, regardless of health insurance status. There have been systematic reviews of both quantitative and qualitative data on parents experiences of NICU admission. The unique aspect of the paper is how these universal themes are manifest in Ghana.

General comments:

Additional context is needed about the health system in Ghana to appreciate the study results. Readers may not understand that families play a critical role in treatment plans for both hospitalized mothers and newborns by purchasing prescribed drugs separately and delivering them to the bedside (at great financial burden for low income families), and physically obtaining and delivering laboratory results to doctors ( increasing both psychological and physical stress on postpartum women). Additional information is also needed about the average income of families in US dollars (since the authors use this conversion metric) so that reader from HICs can understand the financial burden of preterm birth in this LMIC. Issues of insurance of coverage and healthcare cost for ill and preterm/LBW newborns in Ghana have previously been published (Enweronu-Laryea CC, Andoh HD, Frimpong-Barfi A, et al. Parental costs for in-patient neonatal services for perinatal asphyxia and low birth weight in Ghana. PLoS ONE 2018;

555 13: 1–14).

Throughout the manuscript understanding of the maternal experience of care and recommendations for improvement is sometimes stated as “improving clinical care for preterm infants” (line 398). This may be a matter of translation but needs to be corrected; the manuscript needs editing for meaning in English.

Methods:

Participants are labelled as a convenience sample, but the description sounds as if purposive sampling was used (lines 125-126:”…. the aim of achieving varied obstetric history, age and socio-economic characteristics.”). This should be clarified with a description of how screening for representative characteristics was achieved. If a broadly representative sample is achieved should be reported in the results.

The interview asked a leading question concerning finances (Did you have any issue that bothered you, like financial problems, marital problems, job issues or living conditions?) which may have influenced the results, but I think there were other open ended questions that gave participants an opportunity to expand on the issue of financial burden.

Content analysis was appropriate.

Results:

Results are straightforward and illustrate some of the unique stressors families face after preterm birth. The finding that women are not allowed to leave the hospital until the bill is paid even though medically cleared has not been previously reported that I have seen. This was certainly my experience with pediatric patients in Kenya, too.

Discussion:

The Discussion is the strongest part of the paper but would benefit from more context about the health system in Ghana presented earlier in the paper.

While I find these data fascinating, I am not sure that they are unique enough to warrant publication.

6. PLOS authors have the option to publish the peer review history of their article (what does this mean?). If published, this will include your full peer review and any attached files.

**Do you want your identity to be public for this peer review?** For information about this choice, including consent withdrawal, please see our Privacy Policy.

Reviewer #1: No

Reviewer #2: No

---

## [Decision Letter · Decision Letter 1]

31 Oct 2022

Women’s lived experiences of preterm birth and neonatal care for premature infants at a tertiary hospital in Ghana: A qualitative study

PGPH-D-22-00541R1

Dear Dr. Adu-Bonsaffoh,

We are pleased to inform you that your manuscript 'Women’s lived experiences of preterm birth and neonatal care for premature infants at a tertiary hospital in Ghana: A qualitative study' has been provisionally accepted for publication in PLOS Global Public Health.

Best regards,

Rachel Hall-Clifford

Academic Editor

Reviewer Comments (if any, and for reference):

Reviewer's Responses to Questions

**Comments to the Author**

1. If the authors have adequately addressed your comments raised in a previous round of review and you feel that this manuscript is now acceptable for publication, you may indicate that here to bypass the “Comments to the Author” section, enter your conflict of interest statement in the “Confidential to Editor” section, and submit your "Accept" recommendation.

Reviewer #1: All comments have been addressed

Reviewer #2: All comments have been addressed

2. Does this manuscript meet PLOS Global Public Health’s publication criteria? Is the manuscript technically sound, and do the data support the conclusions? The manuscript must describe methodologically and ethically rigorous research with conclusions that are appropriately drawn based on the data presented.

Reviewer #1: Yes

Reviewer #2: (No Response)

3. Has the statistical analysis been performed appropriately and rigorously?

Reviewer #1: Yes

Reviewer #2: (No Response)

4. Have the authors made all data underlying the findings in their manuscript fully available (please refer to the Data Availability Statement at the start of the manuscript PDF file)?

Reviewer #1: Yes

Reviewer #2: (No Response)

5. Is the manuscript presented in an intelligible fashion and written in standard English?

Reviewer #1: Yes

Reviewer #2: (No Response)

6. Review Comments to the Author

Reviewer #1: This is a good paper addressing a gap in maternal and newborn care. The authors have addressed all the points raised and can be published as it is. However, a few typos, omissions, errors must be corrected before final publication.

On line 125 the was used instead of that

On line 126, with was omitted before stillbirth

on line 225, to was omitted before leave

and on line 454, Focused was used instead of focus.

Reviewer #2: (No Response)

7. PLOS authors have the option to publish the peer review history of their article (what does this mean?). If published, this will include your full peer review and any attached files.

**Do you want your identity to be public for this peer review?** For information about this choice, including consent withdrawal, please see our Privacy Policy.

Reviewer #1: No

Reviewer #2: No
